# Proanthocyanidins Alleviate Cadmium Stress in Industrial Hemp (*Cannabis sativa* L.)

**DOI:** 10.3390/plants11182364

**Published:** 2022-09-10

**Authors:** Ming Yin, Langlang Pan, Junfei Liu, Xiaojuan Yang, Huijuan Tang, Yuxin Zhou, Siqi Huang, Gen Pan

**Affiliations:** 1Institute of Bast Fiber Crops, Chinese Academy of Agricultural Sciences, Changsha 410205, China; 2Institute of Crop Sciences, Chinese Academy of Agricultural Sciences, Beijing 100000, China; 3College of Forestry and Biotechnology, Zhejiang A&F University, Hanzhou 311300, China

**Keywords:** hemp, Cd^2+^ stress, proanthocyanidins, transcriptome, metabolome

## Abstract

Industrial hemp (*Cannabis sativa* L.), an annual herbaceous cash crop, is widely used for the remediation of heavy metal-contaminated soils due to its short growth cycle, high tolerance, high biomass, and lack of susceptibility to transfer heavy metals into the human food chain. In this study, a significant increase in proanthocyanidins was found in Yunnan hemp no. 1 after cadmium stress. Proanthocyanidins are presumed to be a key secondary metabolite for cadmium stress mitigation. Therefore, to investigate the effect of proanthocyanidins on industrial hemp under cadmium stress, four experimental treatments were set up: normal environment, cadmium stress, proanthocyanidin treatment, and cadmium stress after pretreatment with proanthocyanidins. The phenotypes from the different treatments were compared. The experimental results showed that pretreatment with proanthocyanidins significantly alleviated cadmium toxicity in industrial hemp. The transcriptome and metabolome of industrial hemp were evaluated in the different treatments. Proanthocyanidin treatment and cadmium stress in industrial hemp mainly affected gene expression in metabolic pathways associated with glutathione metabolism, phenylpropanoids, and photosynthesis, which in turn altered the metabolite content in metabolic pathways of phenylalanine, vitamin metabolism, and carotenoid synthesis. The combined transcriptomic and metabolomic analysis revealed that proanthocyanidins mitigated cadmium toxicity by enhancing photosynthesis, secondary metabolite synthesis, and antioxidant synthesis. In addition, exogenous proanthocyanidins and cadmium ions acted simultaneously on *EDS1* to induce the production of large amounts of salicylic acid in the plant. Finally, overexpression of *CsANR* and *CsLAR*, key genes for proanthocyanidins synthesis in industrial hemp, was established in *Arabidopsis* plants. The corresponding plants were subjected to cadmium stress, and the results showed that *CsLAR* transgenic plants were more tolerant to cadmium than the *CsANR* transgenic and wild-type Arabidopsis plants. The results showed that salicylic acid and jasmonic acid were increased in Arabidopsis overexpressing *CsLAR* compared to AT wild-type Arabidopsis, and levels of secondary metabolites were significantly higher in Arabidopsis overexpressing *CsLAR* than in AT wild-type Arabidopsis. These results revealed how proanthocyanidins alleviated cadmium stress and laid the foundation for breeding industrial hemp varieties with higher levels of proanthocyanidins and greater tolerance.

## 1. Introduction

Industrial hemp (*Cannabis sativa* L.), an annual herb [1], is cannabis with a tetrahydrocannabinol content of less than 0.3% [2] and is not used in drug exploitation. Industrial hemp is used in several countries and regions for heavy metal soil remediation due to its high biomass, large root system, and carbon aggregation capacity to absorb many heavy metal elements [3]. Cadmium (Cd) is a High toxicity and persistent heavy metal that can seriously affect and contaminate soil and agriculture. When plants take up Cd from the soil, this has a detrimental effect on morphology, structure, and physiology. These adverse effects will impede plant growth and development, alter chloroplast ultra-structure, and reduce the efficiency of photosynthesis [4,5,6]. In crop plants, the toxicity of Cd increases oxidative damage, disrupts plant metabolism, and inhibits plant morphology and physiology [7]. In addition, cadmium toxicity affects plants by inhibiting the fixation of carbon and decreasing chlorophyll content and photosynthetic activity [8]. In research on the mitigation of cadmium stress in plants, it was indicated that chemical, plant, and biological governance such as biochar, growth regulators, plant extracts, and rhizobia can be used to improve plant tolerance to cadmium toxicity [7,9,10]. For example, black cumin (*Nigella sativa* L.) seed extracts sprayed on maize leaves can mitigate the adverse impacts of Cd toxicity on plants through its rich antioxidants [11]. There are also plant growth-promoting *rhizo-bacteria* (PGPR), which also reduce the harmful effects of Cd contamination through chelation and improve the growth of wheat [12].

In a previous experiment, when our group used six types of industrial hemp to remediate cadmium-contaminated arable land, we found that Yunnan hemp no. 1 was the most resistant and had the highest yield. The proanthocyanidin content of Yunnan hemp no. 1 increased from less than 0.1 to 5.6 mg/kg under cadmium stress during the seedling stage. Therefore, in this study, we hypothesized that Yunnan hemp no. 1 mitigated cadmium toxicity by up-regulating proanthocyanidin content.

Proanthocyanidins, a plant polyphenol secondary metabolite, has good free radical scavenging ability and antioxidant activity [13,14]. Studies on exogenous proanthocyanidins to alleviate cadmium stress have mostly been reported in animals; relevant studies on plants are scarce and the mechanism has not been elucidated. In recent years, with the development of high-throughput sequencing technology, transcriptome sequencing and corresponding processes have become more and more mature, and are mainly used to study differential gene expression by determining the structure of genes and transcripts, changes in expression under different spatial and temporal conditions, and to mine new functional genes [15,16]. Abiotic stress traits in plants, such as salt tolerance, heavy metal tolerance, and drought tolerance, are controlled by multiple genes, and studying individual genes may lead to inaccurate results; therefore, transcriptomics can provide a more comprehensive analysis of expression at the gene level and elucidate the molecular mechanisms involved [17,18]. The metabolome refers to the overall changes in the endogenous metabolites of an organism, whereas metabonomics is the sum of the dynamic changes caused by external influences on the organism [19,20]. Transcriptomics and metabonomics are well suited to the study of plant responses to abiotic stresses, and combining them for consolidation analysis will help us gain a more comprehensive understanding of the mechanisms of plant stress resistance, which is important for systematic and in-depth exploration of plant stress resistance [21,22]. The process of proanthocyanidin synthesis involves two specific pathways and contains two key rate-limiting enzymes, the Leucoanthocyanidin gene (*LAR*) and Anthocyanin gene (*ANR*) [23]. We performed transgenic overexpression to verify their functions.

In this study, we used Yunnan hemp no. 1 as an experimental material to determine how proanthocyanidins alleviated Cd stress of industrial hemp through phenotypic determination, physiological and biochemical experiments, transcriptome analysis, metabolome analysis, and functional verification of key proanthocyanidin genes.

## 2. Results and Analysis

### 2.1. Analysis of Physiological Indicators in Hemp Grown in Different Experimental Treatments

#### 2.1.1. Effects of Different Treatment Conditions on the Growth of Hemp

After 10 days of growth, Yunnan hemp no. 1 plants were subjected to different treatments. After 3 days, the morphology of the plants had changed significantly and was compared. As can be seen in Figure 1, Yunnan hemp no. 1 grew best using normal treatment, whereas the plants in the other three treatments had varying degrees of growth inhibition in the form of shrinking leaf size and reduced numbers, yellowing, and curling, with correspondingly thinner stalks and shorter and less numerous root systems. The plants in the cadmium treatment group were the most affected, with curled and yellowed leaves and overall wilting. The leaves in the proanthocyanidin treatment group showed obvious leaf yellowing and curling, but they were smaller and were fewer than in the normal group, indicating that their growth was also somewhat inhibited. The proanthocyanidin pretreatment group grew better than the cadmium treatment group.

As can be seen, the proanthocyanidin pretreatment group showed 3.4 cm, 3.25 g, and 0.29 g higher figures for plant height, fresh weight, and dry weight than the cadmium stress group, respectively, thus showing that proanthocyanidin pretreatment could alleviate the damage of cadmium stress on plant growth (Figure 1).

#### 2.1.2. Effects of Different Treatments on the Antioxidant Content of Hemp

Under cadmium stress, plants produce antioxidant substances, which are mainly divided into nonenzymatic antioxidant substances and antioxidant enzymes. To determine the changes in the antioxidant capacity of Yunnan hemp no. 1 under different conditions, after 15 days of growth, plants of uniform length were subjected to four different treatments, and the glutathione (GSH) content and superoxide dismutase (SOD) activity of Yunnan hemp no. 1 leaves were measured at 12, 24, 48, and 72 h of treatment to compare trend changes in antioxidant capacity under different conditions. As shown in Figure 2 with the increase in time, the GSH content in the leaves of each group showed a trend of first increasing and then decreasing, and the difference changed significantly. After 72 h of treatment, the GSH content of each group began to show a decreasing trend. In the first 48 h, both the cadmium and the proanthocyanidin treatment groups produced a large amount of GSH in the industrial hemp plants, with the greatest change in GSH content in the cadmium treatment group and a smaller change in the proanthocyanidin treatment group. In the 48–72 h period, the GSH content of the leaves of all groups began to decrease, and the GSH content of industrial hemp leaves in the proanthocyanidin pretreatment group decreased to a lesser extent than that of the cadmium treatment group, with the difference reaching significance, indicating that proanthocyanidins could alleviate the degree of decrease in nonenzymatic antioxidant substances caused by cadmium stress at the later stages of treatment. As shown in Figure 3, the change in SOD activity was similar to that of the GSH content, with an increasing trend followed by a decreasing trend. At 48 h of treatment, the SOD activity started to show a decreasing trend, and the SOD activity of each group increased by about 2.20%, −7.71%, and −1.55%, respectively. The above data indicated that the trend of the antioxidant enzyme SOD was consistent with the above trend of GSH, which also indicated that proanthocyanidins could alleviate the decline in antioxidant substances caused by cadmium stress at the later stage of treatment.

### 2.2. Multi-Omics Analysis of Hemp Grown in Different Treatment Settings

#### 2.2.1. Transcriptome Results Analysis

The determination of the base distribution and calculation of quality statistics were performed on the data obtained from sequencing. After determination of gene expression using RSEM software (RSEM v1.3.3, Bo Li and Colin N Dewey, Madison, WI, USA), intersample Venn and principal component analysis were performed for all expressed genes based on the expression matrix. Differential gene analysis of transcriptome data using DESeq2 software (DESeq2_1.36.0.; Michael I Love, Wolfgang Huber, and Simon Anders; European Molecular, Biology Laboratory, Heidelberg, Germany) identified 4341 differential genes in the Y_CK_vs_Y_Cd group (2339 up-regulated and 2002 down-regulated), 497 in the Y_CK_vs_Y_Gsp group (250 up-regulated and 247 down-regulated), and 1102 in the Y_Cd_vs_Y_G_Cd group (248 up-regulated and 854 down-regulated) (Figure 4). The number and distribution of these differential genes indicated that external cadmium stress had a large impact on industrial hemp, but most of the genes that were up-regulated due to cadmium stress were down-regulated by proanthocyanidin pretreatment.

GO enrichment analysis was performed for the up-regulated and down-regulated genes (Figure 5a,b). The GO enrich analysis of Y_G_Cd resulted in enrichment mainly in the electron transport chain, protein metabolic process, chlorophyll binding, electron transfer activity, photosystem I, photosystem II, heme binding, protein modification process, cellular protein modification process, protein–chromophore linkage, oxidation-reduction process, and photosynthetic electron transport chain. The results of analysis of down-regulated genes were mainly in the cellular-modified amino acid biosynthetic process, tetrahydrofolate metabolic process, pteridine-containing compound biosynthetic process, folic acid-containing compound biosynthetic process, cytochrome complex assembly, mitochondrial respiratory chain complex assembly, tetrahydrofolate biosynthetic process, carbon fixation, respiratory chain complex IV assembly, and mitochondrial cytochrome oxidase assembly.

In the GO enrichment analysis of the Y_CK_vs_Y_Cd group, differential genes were mainly enriched in metabolic pathways such as microtubule synthesis, photosynthesis, nucleosomes, protein–DNA complex, photosystem, cellular amino acid catabolic processes, and DNA packaging complex. In the Y_Cd_vs_Y_G_Cd group, the differential genes were mainly enriched in metabolic pathways such as oxidoreductase activity, hemoglobin binding, tetrapyrrole binding, iron ion binding, protein–chromatin particle attachment, chlorophyll binding, and the MCM complex (Figure 5c).

KEGG enrichment analysis showed that in the Y_CK_vs_Y_Cd group, differential genes were mainly enriched in metabolic pathways such as tyrosine metabolism, flavonoid biosynthesis, phytohormone signaling, phenylalanine metabolism, glutathione metabolism, photosynthesis, and phenyl propane biosynthesis (Figure 6). The data indicated that Cd^2+^ affected the production of secondary metabolites, phytohormone production, and photosynthetic responses in plants. In the Y_Cd_vs_Y_G_Cd group, differential genes mainly affected metabolic pathways such as phenyl propane biosynthesis, carbon fixation in photosynthetic organisms, amino acid and nucleotide sugar metabolism, and linolenic acid synthesis. The data suggested that pretreatment with proanthocyanidins affected photosynthesis, sugar metabolism, and linolenic acid metabolism.

The differentially expressed genes with high expression of abscisic acid (*ABA*, two genes), ethylene (*ETH*, three genes), salicylic acid (*SA*, one gene), and jasmonic acid (*JA*, two genes) were selected from among the differentially expressed genes for comparative expression analysis, and these genes were found to act in response to stress in plants, resulting in the production of a large number of corresponding hormones(Figure 7a). In the cadmium stress group, all genes were up-regulated to varying degrees compared to the normal environment. In contrast to the cadmium stress group, two genes were up-regulated and six genes were down-regulated in the proanthocyanidin pretreatment group. This indicated that plants up-regulated the expression of most phytohormone-related genes in response to stress, whereas proanthocyanidins decreased the expression of most phytohormone genes, and that the proanthocyanidin pretreatment group reduced the expression of phytohormone-related genes in response to cadmium stress.

When Cd^2+^ is absorbed from the plant root system into the plant, it causes increased oxidative damage to the plant, which results in the up-regulation of antioxidant substances and transporter proteins [24]. From the differentially expressed genes, peroxidase (*POD*, two genes), ascorbic acid (*ASA*, vitamin C, two genes), glutathione (*GSH*, two genes), (*SOD* two genes), and hydrogen peroxide (*CAT*, catalase, two genes) were analyzed in comparison with each other (Figure 7b). In comparison to the cadmium stress group, in the proanthocyanidin pretreatment group, five genes were up-regulated and five genes were down-regulated. These data indicated that the five up-regulated genes were the key genes involved in the proanthocyanidin-mediated alleviation of cadmium stress.

Finally, plants produce a large number of secondary metabolites to mitigate the damage caused by external stress. Differential genes from the transcriptome of the three largest gene families associated with secondary metabolites in plants, *MYB*, *WRKY*, and *NAC* [25,26,27], were selected for comparison with three genes selected from each gene family (Figure 7c). The genes were differentially up- or down-regulated in all treatment groups compared to the normal environment. In contrast, six genes were down-regulated, and two genes were up-regulated in the proanthocyanidin pretreatment group when compared to the cadmium stress group. Among them, the expression trends of *LOC115718987* and *LOC115706270* genes in the proanthocyanidin pretreatment group compared to the cadmium stress group were not consistent with the trends of both the cadmium stress group and proanthocyanidin group compared to the normal environment group, suggesting that these two genes may be involved in the proanthocyanidins-mediated alleviation of cadmium stress.

#### 2.2.2. Metabolome Results Analysis

Industrial hemp seedlings with the same transcriptome and consistent growth were selected for metabolomics assays, and after determining the final metabolites, the screened differential metabolites were subjected to KEGG annotation and metabolic pathway analysis. Significantly different metabolites were identified using Student’s *t*-test and the OPLS-DA model, with the screening criteria being a *p*-value less than 0.05 and a model first principal component importance (VIP) greater than 1 [28]. Figure 8 shows that 28 differential metabolites (12 down-regulated and 16 up-regulated) were identified in the CK_vs_Cd group, eight differential metabolites (three up-regulated and five down-regulated) in CK_vs_Gsp, and 13 differential metabolites (three up-regulated and 10 down-regulated) in Cd_vs_Cd+Gsp. The above differential metabolites were compared to obtain Venn plots for each group (Figure 8). Among them, sixteen differential metabolites (43.2%) were specific to the CK_vs_Cd group, two (5.4%) were specific to the CK_vs_Gsp group, seven (18.9%) were specific to the Cd_vs_Cd+Gsp group, six (16.2%) were common to the CK_vs_Cd group and the CK_vs_Gsp group, six (16.2%) were common to the CK_vs_Cd group, and six (16.2%) were common to the Cd_vs_Gsp+Cd group, which was also generally consistent with the distribution of differential genes in the transcriptome.

After differential metabolites were obtained, the differential metabolites were annotated using the KEGG database and further metabolic pathway enrichment analysis and topological analysis of the differential metabolites were performed [29]. Figure 9 shows that the CK_vs_Cd group was found to have significantly decreased riboflavin (vitamin B2) metabolism, nicotinate and nicotinamide metabolism (vitamin B3), and purine metabolism (guanosine). Phenylalanine (salicylic acid) metabolism, indole biosynthesis of tyrosine and tryptophan (alkaloids), cysteine and methionine metabolism with arginine and proline metabolism (S-adenosylmethionine), and carotenoid biosynthesis (abscisic acid) increased. In the Cd_vs_Cd+Gsp group, the pentose phosphate pathway (gluconic acid), biosynthesis of carotenoids (abscisic acid), and biosynthesis of phenyl propanoids (coumarin) decreased, and metabolism of phenylalanine (salicylic acid) increased. After obtaining matching information for each group of contrasting differential metabolites, we performed a pathway search and regulatory interaction network analysis with the KEGG database using the corresponding species of Arabidopsis. In the CK_vs_Cd group, riboflavin metabolism and purine metabolism were significantly down-regulated, and salicylate metabolism was significantly up-regulated. Among them, riboflavin content in the inhibited metabolic pathway was significantly decreased and linked to purine metabolism via *FAD* diphosphatase and ribulose diphosphatase. Guanosine content in purine metabolism was decreased and affected the synthesis of nicotinic acid and S-adenosyl-L-methionine via guanosine ribose hydrolase and phosphoribosyltransferase. The levels of abscisic acid and salicylic acid in the stimulated metabolic pathways rose, with indole in turn affecting dihydroxybenzoate via L-serine hydrolase, both of which rose. In the Cd_vs_Gsp+Cd group, the phenylalanine metabolic pathway promoted a rise in salicylic acid via the protein serine enzyme, whereas the phenylpropanoid metabolism and pentose phosphate pathway metabolism were significantly down-regulated via β-glucosidase, glucose kinase, and ribulose phosphate diphosphate kinase, promoting a decrease in abscisic acid, coumarin, and gluconic acid.

#### 2.2.3. Combined Analysis of Transcriptome and Metabolome Results

After a plant senses an external environmental stimulus, such as heavy metal stress, drought stress, or salt stress, the plant will change its gene expression in response to the environmental change to reduce the environmental impact on it [30]. The differential genes will lead to the production of a large number of differential metabolites in the plant in response to external stresses. Therefore, the results of transcriptome analysis need to be combined with those of the metabolome analysis to explore the overall changes in plant response to cadmium stress.

In this study, the combined analysis of transcriptomic and metabolomic data in Figure 10 showed that industrial hemp plants produced large amounts of proanthocyanidins when exposed to cadmium stress. When industrial hemp was exposed to cadmium stress, Cd^2+^ may act with proanthocyanidins through calcium-binding proteins on *EDS1*, an important plant immunity gene that induces salicylic acid production and inhibits the production of jasmonic acid in plants [31,32]. This is in line with the secondary metabolites in leaves of industrial hemp after cadmium stress and trends that are consistent with changes in the pathway acting on *EDS1*. The expression trends of *ANR*, *EDS1*, *CBP*, *CBT*, and *LLP* were all similar to the trends of metabolite changes.

The production of salicylic acid in industrial hemp alleviated cadmium toxicity in three main ways. The first is the photosynthetic pathway [33], where the expression of genes such as *PsbO*, *PsbP*, *FNR*, and *ATPC* in the photosynthetic metabolic pathway were all down-regulated under cadmium treatment, indicating that cadmium toxicity affected the photosynthesis of the plant. The expression of these genes was up-regulated in both the proanthocyanidin pretreatment group and proanthocyanidin treatment group, indicating that the substance alleviated the inhibition of the photosynthetic metabolic pathway of industrial hemp by cadmium stress after exogenous application of proanthocyanidins. The second is plant secondary metabolites controlled by the *MYB*, *bHLH*, *NAC*, and *bZIP* gene families [34]. The genes of this pathway were all up-regulated under cadmium stress treatment, and these genes were the key secondary metabolite-related genes for stress alleviation by proanthocyanidin pretreatment, and include *MYBS3*, *MYB78*, and *bHLH11*. The expression trends of *NECD* and *PLA* were consistent with those of the corresponding metabolites abscisic acid and coumarin, indicating that the cadmium toxicity in industrial hemp was alleviated in the proanthocyanidin pretreatment group and that corresponding secondary metabolite levels began to decrease. The third pathway is the antioxidant substance-related pathway [35,36], with a similar expression of genes to the second pathway. Four of these genes, *CAT*, *CAT2*, *SOD*, and *POD31*, were significantly up-regulated in the proanthocyanidin pretreatment group and could be key antioxidant substance-related genes to alleviate cadmium stress. These three pathways constitute a regulatory network for the alleviation of cadmium stress in the exogenous proanthocyanidin pretreatment group in this study.

#### 2.2.4. Validation of Differentially Expressed Genes by qRT-PCR

To verify the reliability of the transcriptome data, five genes were selected, and trends in their expressions were determined in four treatments using the same samples as the transcriptome, based on the functions of the genes annotated in the transcriptome and their expressions (Figure 11). The qPCR revealed that only one differential gene in one group was not consistent with the transcriptome, while the other 14 differential trends were consistent with the transcriptome data, indicating that the transcriptome data were reliable.

### 2.3. Cloning and Functional Analysis of Key Genes for Procyanidin Synthesis of Hemp

#### 2.3.1. Construction of ANR and LAR Overexpression Lines of Hemp

The key genes for proanthocyanidin production in industrial hemp, *ANR* (*LOC115708131*) and *LAR* (*LOC115699323*), were overexpressed transgenically in Arabidopsis, and transgenic T3 generation Arabidopsis seeds were selected for germination experiments at different levels of cadmium stress. Thirty Arabidopsis seeds were sown at each concentration, and three parallels were set up to count seed germination numbers. The number of seeds that germinated was counted to determine whether the transgenic plants were tolerant to cadmium stress. Arabidopsis seedlings were taken after 10 days of germination and treated with 100 μmol/L of cadmium in water for 4 days to observe phenotypic changes (Figure 11). To select positive transgenic seedlings, T1-generation transgenic Arabidopsis seeds were sown on the medium containing resistance, vernalized for 3 days, and placed in an artificial climate chamber for 10 days. Transgenic positive seedlings (which had rooted and germinated normally and grown true leaves) were selected from the cultured seedlings, and DNA was extracted for PCR identification by thioformycin. It can be seen from Figure 12 that the selected positive seedlings were all transgenic, and transgenic Arabidopsis were selected. All seedlings were determined to be positive, and T1 generation Arabidopsis positive seedlings were collected.

#### 2.3.2. Verification of Stress Tolerance in Transgenic Plants

To verify whether the transgenic Arabidopsis thaliana was resistant to cadmium stress, *CsANR* and *CsLAR* transgenic T3 generation seeds and wild-type (AT) seeds were grown hydroponically with cadmium treatment. Figure 13 showed that *CsLAR*-transformed Arabidopsis thaliana grew best, with more and larger leaves and larger and longer root systems; wild-type and *CsANR*-transformed plants had yellowed leaves and shorter and fewer root systems. The growth physiological data of *CsLAR*-transformed Arabidopsis thaliana were significantly better than those of *CsANR*-transformed plants and wild-type Arabidopsis in terms of plant height, fresh weight, and dry weight. The *CsLAR*-transformed Arabidopsis thaliana showed 23.5 mm, 0.049 g, and 0.003 g higher figures for plant height, fresh weight, and dry weight, respectively, than wild-type Arabidopsis. The results showed that *CsLAR* significantly improved the resistance of Arabidopsis to external cadmium (Figure 13).

#### 2.3.3. Determination of Secondary Metabolites in Transgenic Plants

The phytohormone contents of AT wild-type Arabidopsis and overexpressed Arabidopsis *ANR* (*LOC115708131*) and *LAR* (*LOC115699323*) were determined and analyzed by ultra-performance liquid chromatography (Vanquish, UPLC, Thermo, Thermo Fisher, Waltham, MA, USA) and high-resolution mass spectrometry (QExactive, Thermo, USA) (Figure 14). For the phytohormones, salicylic acid, 3-indoleacetic acid, jasmonic acid, jasmonic acid-isoleucine, deoxyribonucleic acid, erythromycin A3, erythromycin A1, erythromycin A4, and erythromycin A7 were measured, and 0.2 mL of each sample was taken. The differences in their phytohormone contents were concentrated in the four categories of salicylic acid, 3-indoleacetic acid, deoxyribonucleic acid, and jasmonic acid. The differences were 5.71 ng, −0.917 ng, −0.412 ng, and 6.226 ng, respectively. The results indicated that salicylic acid and jasmonic acid were elevated in overexpressing Arabidopsis compared to wild-type AT Arabidopsis, suggesting consistency with the results of the combined transcriptome and metabolome analyses. Their plants were able to produce more salicylic acid to enhance the tolerance of overexpression plants to cadmium stress.

After obtaining significant alterations in the phytohormones of overexpressed Arabidopsis thaliana, secondary metabolites of the plants were extracted and analyzed for changes in their content. From the analysis of the content of the differential metabolites, we learned that the changes in anthocyanin content were similar, with a significant increase in proanthocyanidins (Figure 15). In particular, the most significant changes in proanthocyanidin content were obtained from the AT vs CsLAR assay, with changes in the concentrations of kaempferol-3-*O*-rutinoside, naringenin, and naringenin-7-*O*-glucoside. Compared with AT, the contents of these four categories increased by 0.67779491, 0.002151478, 1.69283349, and 0.35842076, respectively. The changes in proanthocyanidin content and salicylic acid content were generally consistent with the transcriptomic and metabolomic data. It is sufficient to demonstrate that *LAR* and *ANR* are key genes affecting proanthocyanidin synthesis in plants, and their gene functions enable the network regulation of proanthocyanidin mitigation by complementing proanthocyanidins to increased proanthocyanidin content within plants.

## 3. Discussion

### 3.1. Effects of Different Treatments on the Physiological Indicators of Industrial Hemp

The external stress environment causes the plant to produce large amounts of re-active oxygen species [31]. In response to oxidative damage, the plant produces large amounts of antioxidant substances internally [37]. As in Section 2.2.2 and Section 2.2.3, all groups of industrial hemp plants produced large amounts of GSH and SOD in the first 48 h. As the treatment time increased, Cd^2+^ disrupted the ability of the plant to synthesize antioxidant substances [38]. For instance, in the 48–72 h period after treatment, the antioxidant substances inside the plants of industrial hemp began to decrease. By comparing trends in the phenotypic and antioxidant content of industrial hemp plants in different treatments, it was determined that both cadmium, a heavy metal, and proanthocyanidins are environmental stressors in industrial hemp, but pretreatment with proanthocyanidins was effective in alleviating cadmium toxicity. The pretreatment of proanthocyanidins also improved plant tolerance to Cu stress, including recovery of plant growth and lignin synthesis [39]. It was shown that pretreatment of proanthocyanidins could induce the establishment of plant defense mechanisms in response to heavy metal cadmium, which allowed industrial hemp to prevent the decreasing trend of antioxidant substances in plant leaves at a late stage of exposure to Cd^2+^.

### 3.2. Changes in Gene Expression and Metabolite Content of Hemp under Different Treatments

Upon sensing external environmental stimuli, such as heavy metals, drought, or salt, plants alter gene expression to reduce their environmental impact [30]. Through transcriptome assays of industrial hemp grown in different conditions, 4341 differential genes (2339 up-regulated and 2002 down-regulated) were identified in the Y_CK_vs_Y_Cd group, and there was a high number of differential genes in this fraction that were largely in the same proportion of up and down-regulation, implying that cadmium treatment had the greatest effect on industrial hemp. The GO enrichment results for this fraction of genes were mainly focused on microtubules, photosynthesis, nucleosomes, and protein-DNA complexes, indicating that cadmium stress mainly affected nucleic acid production, amino acid synthesis, and protein synthesis [40]. The KEGG enrichment results for these genes focused on synthetic pathways such as flavonoid biosynthesis, phytohormone signaling, phenylalanine metabolism, and glutathione metabolism, all of which are key metabolic pathways for plants to cope with external stress, and suggest that industrial hemp produces a large number of secondary metabolites in response to external cadmium stress stimuli. In the Y_Cd_vs_Y_G_Cd group, with 1102 differential genes (248 up-regulated and 854 down-regulated), most of the genes were down-regulated, corresponding to genes down-regulated by proanthocyanidins to alleviate cadmium stress. The GO enrichment results for differential genes in this group focused on oxidoreductase activity, tetrapyrrole binding, chlorophyll binding, and the MCM complex, all of which are involved in the way plants respond to oxidative damage DNA repair [41,42]. The enrichment results for KEGG also focused on phenyl propane biosynthesis, carbon fixation in photosynthetic organisms, aminosaccharide and nucleotide sugar metabolism, and also showed that pretreated proanthocyanidins alleviated the cadmium stress in industrial hemp in terms of secondary metabolites, ATP synthesis, and proteins and nucleic acids.

Comprehensive analyses of transcriptomic and metabolomic data showed that industrial hemp plants produced large amounts of proanthocyanidins when exposed to cadmium stress. When industrial hemp was exposed to cadmium stress, Cd^2+^ may act with proanthocyanidins through calcium-binding proteins on *EDS1*, an important plant immunity gene that induces salicylic acid production and inhibits jasmonic acid production [31,32], which is in line with the secondary metabolites found in leaves after cadmium stress. Salicylic acid is an important phytohormone for plant resistance to external stresses. Industrial hemp under proanthocyanidin treatment also produced significant amounts of gentianic acid [43], a secondary derivative of salicylic acid, suggesting that proanthocyanidins had an indirect association with salicylic acid. Salicylic acid activates the defense of poplar against biotrophic rust fungi by increasing the biosynthesis of catechins and Proanthocyanidins [44]. In the *EDS1* gene pathway, the expression trends of ANR, EDS1, CBP, CBT, and LLP were all similar to those of the metabolites.

Industrial hemp produces salicylic acid, which mitigates cadmium toxicity in three main ways. The first is the photosynthetic pathway, where cadmium stress is photosynthesis-promoting at low concentrations and photosynthesis-inhibiting at high concentrations [33]. The second pathway is the secondary plant metabolism controlled by *MYB*, *bHLH*, *NAC*, *bZIP*, and other gene families. The third pathway is the antioxidant substance-related pathway, and the expression of genes associated with this pathway is similar to that of the second pathway. Four of these genes, *CAT*, *CAT2*, *SOD,* and *POD31*, were significantly up-regulated in the proanthocyanidin pretreatment group and may be key antioxidant substance-related genes to alleviate cadmium stress in industrial hemp, whereas the rest of the genes may not be key genes. These three pathways, however, constitute the regulatory network of exogenous proanthocyanidin pretreatment for the alleviation of cadmium stress.

### 3.3. Construction and Functional Analysis of ANR and LAR Overexpression in Plants

Plants produce large amounts of endogenous proanthocyanidins in response to cadmium stress [45]. To investigate the function of their production, we used trends in the antioxidant substances produced by exogenous proanthocyanidins to alleviate cadmium stress, transcriptomic data, and metabolomic data to probe the induction of salicylic acid. Based on the germination rate and growth of transgenic plants under different cadmium stresses, it was determined that Arabidopsis thaliana plants transgenic to the *CsLAR* gene improved the ability to cope with cadmium stress. *LAR* and *ANR* are involved in proanthocyanidins biosynthesis in apple [46], and overexpression of *ANR* enhances tolerance to abiotic stress in tobacco by increasing the regulation of *ROS* scavenging and *ABA* signaling [47]. In subsequent measurements of phytohormones and secondary metabolites, salicylic acid and jasmonic acid content and proanthocyanidin content in secondary metabolites increased significantly in Arabidopsis plants transgenic for the *CsLAR* gene. It is clear from this result that when overexpressed, *LAR* and *ANR* reduced oxidative damage and eliminated free radicals in plants by increasing salicylic acid content in vivo under a cadmium stress environment. The *LAR* and *ANR* genes induced the production of endogenous proanthocyanidins in the plants, which further improved their tolerance to cadmium stress. The results are in agreement with the above-mentioned discussion, which confirmed that both *CsANR* and *CsLAR* were key genes for the production of proanthocyanidins in industrial hemp.

## 4. Materials and Methods

### 4.1. Plant Materials and Growth Conditions

Yunnan hemp no. 1 (YM1) was selected as the experimental material and preserved by the annual hemp crop genetic improvement innovation team at the Institute of Hemp Research, Chinese Academy of Agricultural Sciences, China. The Arabidopsis thaliana used for overexpression studies was selected from Columbia wild-type Arabidopsis thaliana. Environmental settings were diurnal temperature of 26/20 °C, photoperiod of 16/8 h (light/dark), relative humidity of 60%, and light intensity of 700 µmol m^−2^, and the nutrient solution was changed once every two days.

### 4.2. Stress Treatment and Determination of Antioxidant Substances

After 10 days of hemp growth, seedlings of uniform growth were selected for four different treatments: (1) control treatment for 4 days (nutrient solution only); (2) cadmium stress treatment for 4 days (nutrient solution containing 100 µmol/LCdCl2); (3) proanthocyanidin treatment for 4 days (nutrient solution containing 50 mg/L proanthocyanidin concentration); (4) proanthocyanidin pretreatment for 12 h (same treatment as step 3), followed by cadmium treatment for 4 days (same treatment as step 2). Six biological replicates were set up for each group, and plants of uniform growth were selected for subsequent trials.

To conduct the *GSH* and *SOD* assay methods, a homogenate (10%) was prepared as follows. Plant tissue assay site tissue was weighed (g), combined with PBS (phosphate-buffered saline) buffer (mL; 1:9), and homogenized. The homogenate was centrifuged and the supernatant was taken for use in the assay. An equal volume of precipitant from a kit was added to the supernatant, mixed well, and centrifuged. The supernatant was used as the sample to be tested. *GSH* and *SOD* were detected according to the instructions using the corresponding detection kit (Nanjing Jiancheng Bioengineering Institute, Nanjing, China), and the *GSH* and *SOD* contents were quantified colorimetrically at 405 nm.

### 4.3. Transcriptome Analysis

After each group of industrial hemp had been treated, three whole industrial hemp seedlings of uniform growth were taken from each treatment environment, total RNA was extracted, and a total RNA mass greater than 2 μg was tested for quality and purity at OD260/OD280 = 1.9–2.0 using NanoDrop 2000 software (Thermo Fisher, Waltham, MA, USA). The samples were assayed by transcriptomics using Illumina HiSeq™ 4000 software (HiSeq™ 4000 v3.4.0, San Diego, CA, USA). The raw data were assessed for relevant quality and quality-filtered to obtain high-quality (mass values > 20) pure reads. The filtered data were compared to the cannabis genome data (GCF_900626175.2) using the software TopHat2 (TopHat2 v.2.1.; Daehwan Kim; Johns Hopkins University, Baltimore City, MA, USA) for assembly annotation and quality assessment of the entire transcriptome. The software StringTie was used to splice the compared data and to mine for new genes while comparing it to the original genomic data. Databases (NR, Swiss-Prot, Pfam, EggNOG, GO, and KEGG) were used for gene annotation and statistics, and RSEM software was used for gene expression calculations. The Ck group was compared with the cadmium group, the Ck group with the Gsp group, and the cadmium group with the Cd+Gsp group for gene expression. The software DESeq2 was used for differential expression analysis. The screened differential genes were subjected to GO and KEGG enrichment analysis and the *p* value was tested using Fisher’s algorithm, and was required to be less than 0.01. The top 10 GO results and KEGG pathways in each group were selected for enrichment, and the enrichment analysis results of the three differential gene groups were pooled to compare their differences.

### 4.4. Metabolome Analysis

Industrial hemp seedlings with the same transcriptome and consistent growth were selected for metabolome determination. Six biological replicates of material from each environment were selected to obtain a total of 24 experimental samples from the four treatments. Metabolites were determined using ultra-performance liquid chromatography-quadrupole time-of-flight mass spectrometry (UHPLC-QTOF MS) [47]. A UPLCHSST3 column (1.8 μm, 2.1 × 100 mm) was used at 40 °C with an injection volume of 2 μL at 4 °C in auto-injection. The high-resolution mass spectrum was ABSciexQTOF (Darren R Allen and Brett C McWhinney;Pathology Queensland Central Laboratory, Heston QLD, Australia ), and the high-resolution mass spectrometry data acquisition was performed using the information-dependent acquisition IDA (information-dependent acquisition) mode. Data analysis of the acquired data was performed using the BiotreeDB database and MAPS software. MRM (Multiple Reaction Monitoring) data acquisition was performed for all samples on the triple quadrupole mass spectrometer by combining the parent ions with the daughter ions in the secondary spectrum to form ion pairs to build an MRM database. Once the metabolites were identified, the final metabolites were screened for significant differences using Student’s *t*-test to determine the final metabolites. The screened differential metabolites were subjected to KEGG annotation and metabolic pathway analysis.

### 4.5. The qRT-PCR Analysis

The RNA extracted as described in Section 2.3.1 was used for cDNA reverse transcription, and the cDNA was used for qPCR validation. Eight genes were selected for validation. Primers were designed using Primer5, and hemp actin was selected as the internal reference gene (sense: CCAATAGCCTTGCATT CCAT; anti-sense: TCGATTGGAAAGCCGAATAC.).

### 4.6. Vector Construction and Genetic Transformation

The annotated cannabis genome was downloaded from the National Center for Biotechnology Information (NCBI) database (nih.gov) to obtain sequence information on the *ANR* and *LAR* genes, key genes for proanthocyanidin production in industrial hemp. Based on the obtained gene sequences, the upstream and downstream sequences were designed using Primer5 software, and the target genes were amplified using the cDNA of Yunnan hemp no. 1. Agarose electrophoresis was performed to obtain the target bands, and the target bands were recovered using a gel recovery kit. The cloning vector was constructed using a pPOTO vector to obtain a large number of cloned genes. After mixing, the plates were left at 37 °C for 15 min, transformed with *E. coli*, coated in a selection medium containing 100 mg/L ampicillin, and single clones were picked.

### 4.7. Measurement of Tolerance in Transgenic Plants

The key genes for proanthocyanidin production in industrial hemp, *ANR* (*LOC115708131*) and *LAR* (*LOC115699323*), were overexpressed transgenically in Arabidopsis, and transgenic T3 generation Arabidopsis seeds were selected for germination experiments at different concentrations of cadmium. Thirty Arabidopsis seeds were sown at each concentration, and three parallels were set up to count the number of germinated seeds and to determine cadmium tolerance in transgenic plants. The number of seeds germinated was counted to determine whether the transgenic plants were tolerant to cadmium stress. After 10 days of germination, Arabidopsis seedlings were treated with 100 μmol/L of cadmium in water for 4 days to observe phenotypic changes.

### 4.8. Measurement of Phytohormones in Transgenic Plants

A sample of industrial hemp leaf (0.1 mg) was ground and 1 mL of ice-cold 50% ACN aqueous solution was added. The sample was sonicated for 3 min at 4 °C, incubated for 30 min at 4 °C, and then centrifuged at 12,000 rpm for 10 min at 4 °C. The supernatant was removed and passed through an RP-SPE column with 1 mL of 100% MeOH and 1 mL of deionized water, then equilibrated with 50% ACN aqueous solution (*v*/*v*). After loading the sample (supernatant obtained according to the above steps), the flow-through graded fraction was collected in a glass tube. The column was then rinsed with 1 mL of 30% ACN (*v*/*v*) and the fraction was collected in the same glass tube as the flow-through fraction. The sample was dried in a stream of nitrogen, dissolved in 200 μL 30% ACN (*v*/*v*), and transferred to a sample vial containing the insert. The data acquisition instrumentation system consisted mainly of UPLC and high-resolution mass spectrometry (QExactive; Thermo Fisher, USA). Electrospray ionization (ESI) was used with a sheath gas of 40 arbs, an auxiliary gas of 10 arbs, an ion spray voltage of −2800 V, a temperature of 350 °C, and an ion transport tube temperature of 320 °C. The scanning mode was single ion detection (SIM) mode and negative ion.

### 4.9. Measurement of Anthocyanins in Transgenic Plants

Three samples were selected for this project: wild-type Arabidopsis (AT) and Arabidopsis overexpressing G1 or G2. The samples were first freeze-dried under vacuum and ground to powder using a ball mill (30 Hz, 1.5 min), and 50 mg of powder was weighed and dissolved in 500 μL of extract (50% aqueous methanol solution containing 0.1% hydrochloric acid). Samples were then vortexed for 5 min, sonicated for 5 min, and centrifuged for 3 min (12,000 rpm, 4 °C). The supernatant was aspirated and the operation was repeated once. The two supernatants were combined, filtered through a microporous membrane (0.22 μm pore size), and stored in an injection bottle for UPLC tandem mass spectrometry (MS/MS) analysis. The data acquisition instrumentation system consisted mainly of UPLC and MS/MS. Qualitative and quantitative MWDB (MetWare Database) databases were constructed based on standards for the qualitative analysis of data from mass spectrometry assays.

### 4.10. Statistical Analysis and Presentation of Totals

Three biological replicates were set up in the industrial hemp cadmium stress phenotype assay, the ***GSH*** non-enzymatic antioxidant assay, and the ***SOD*** enzymatic antioxidant assay. In contrast, three biological replicates and three technical replicates were set up in the overexpression Arabidopsis cadmium stress experiment. After the assay data were obtained, SPSS or other data statistical software was used to obtain the significance relationship between any two means. Different alphabetic letters indicate that the differences between groups of data at the same time point reached significance, *p* < 0.05 (Duncan). Values are expressed as the mean ± standard deviation; *n* = 3.

## 5. Conclusions

In summary, we found that exogenous application of proanthocyanidins could also significantly reduce the impact of Cd stress on industrial hemp. Transcriptomics analysis could significantly alleviate the gene expression up-regulation of some plant hormones, antioxidant enzymes, and some transcription factor families in response to Cd stress. Exogenous proanthocyanidins and Cd^2+^ may act by increasing *CsEDS1* expression to induce salicylic acid production, to finally alleviate Cd stress. The proanthocyanidin key gene, *CsLAR*, could significantly improve the resistance of Arabidopsis to external Cd stress by producing more procyanidins and plant hormones.

## Figures and Tables

**Figure 1 plants-11-02364-f001:**
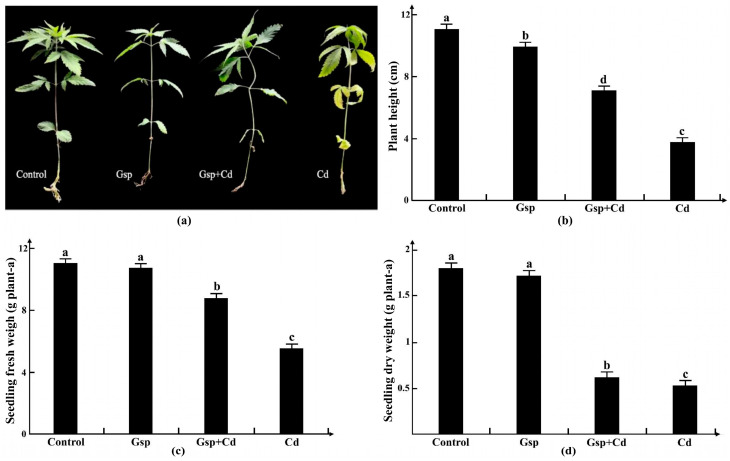
Comparison of hemp morphology in different processing treatments. (**a**) The control represents the normal growth environment group, Gsp represents the proanthocyanidin treatment group, Cd+Gsp represents the cadmium stress group with proanthocyanidin pretreatment, and Cd represents the cadmium stress group. (**b**) Plant height; (**c**) fresh weight; (**d**) dry weight. Different letters indicate significant differences in the data using different treatments, *p* < 0.05 (Duncan). Data are expressed as the means (±SD), *n* = 10.

**Figure 2 plants-11-02364-f002:**
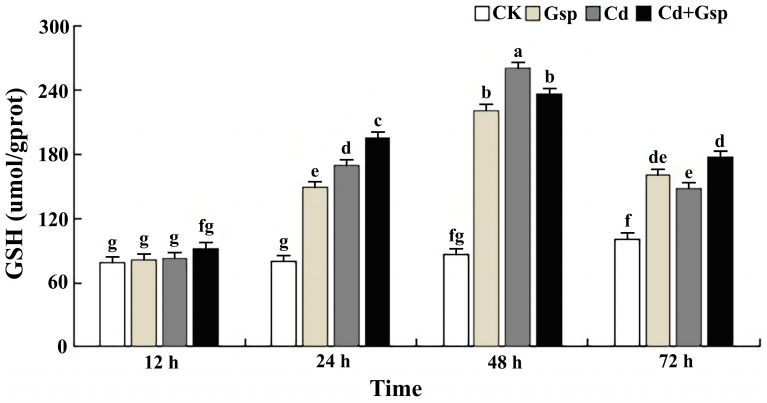
Changing trend in the GSH content of hemp in different treatments. Note: The values are mean ±SD. Different letters indicate significant differences in the data of each group at the same time point, *p* < 0.05 (Duncan). Data are expressed as the means (±SD), *n* = 3.

**Figure 3 plants-11-02364-f003:**
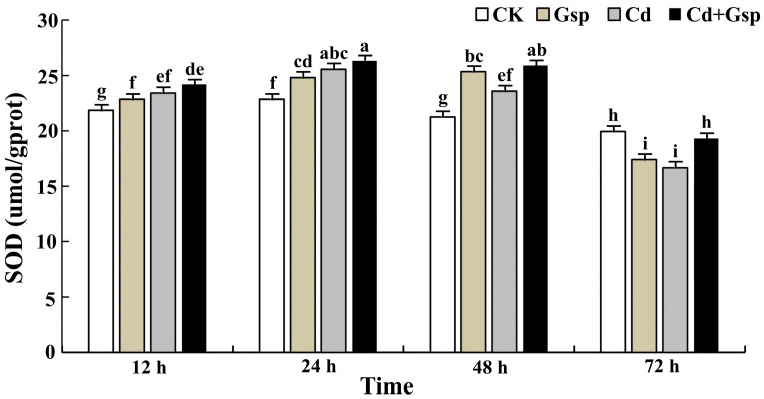
Change trend of SOD activity of hemp in different processing treatments. Different letters indicate significant differences in the data of each group at the same time point, *p* < 0.05 (Duncan). Data are expressed as the means (±SD), *n* = 3.

**Figure 4 plants-11-02364-f004:**
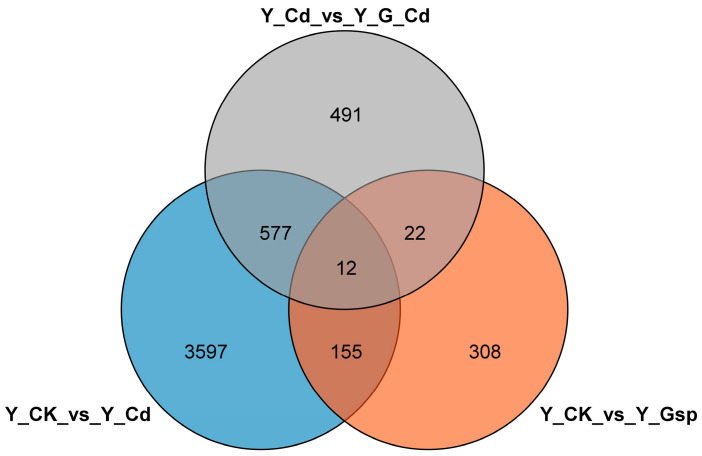
Venn diagram analysis of differentially expressed genes. Note: The different color circles represent different groups of genes, and the intersection of the circles represents the number of genes shared between each gene set.

**Figure 5 plants-11-02364-f005:**
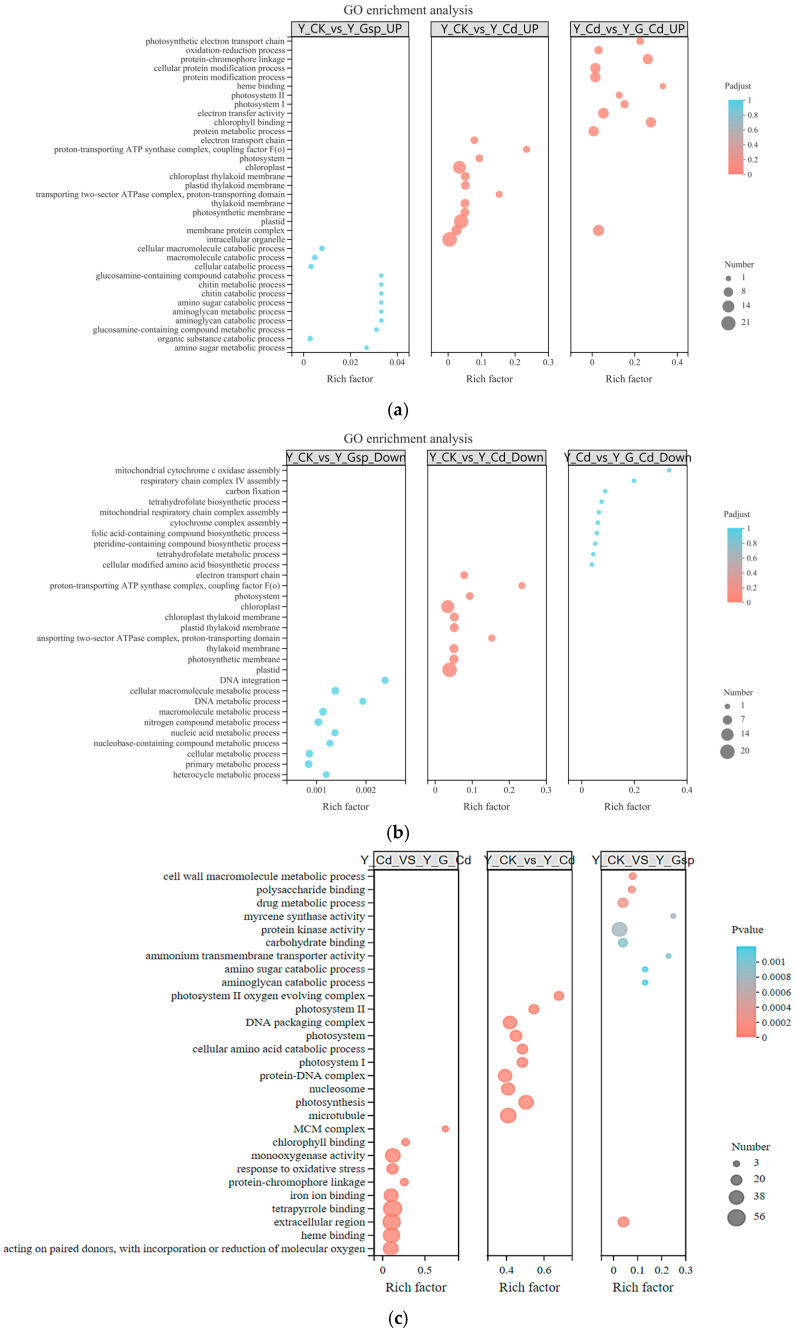
GO enrichment analysis. (**a**) GO enrichment analysis of up-regulated genes; (**b**) GO enrichment analysis of down-regulated genes; (**c**) GO enrichment analysis of differential genes.

**Figure 6 plants-11-02364-f006:**
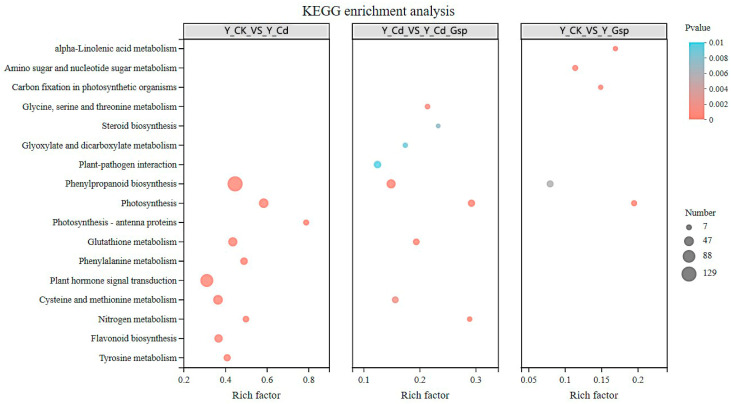
KEGG enrichment analysis.

**Figure 7 plants-11-02364-f007:**
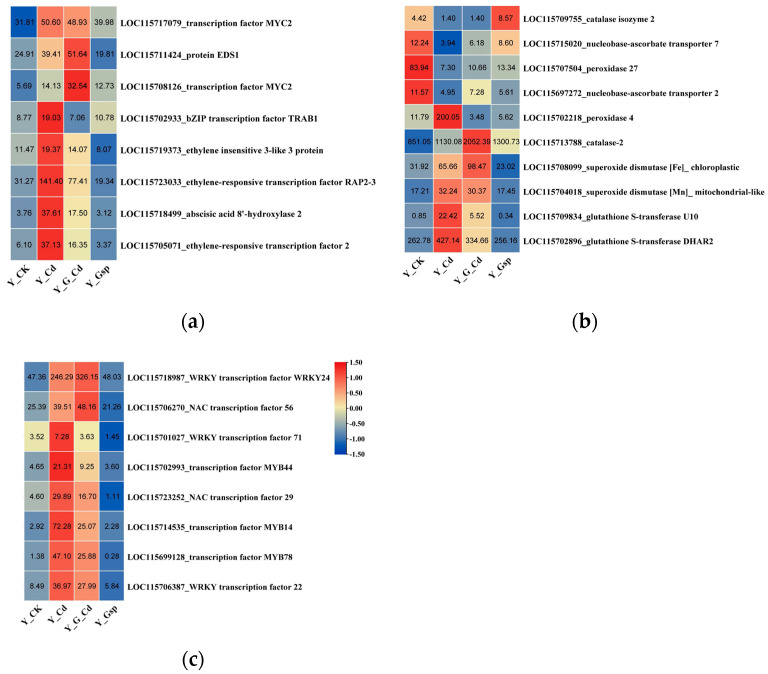
Comparison of expression of differentially expressed genes related to (**a**) phytohormones, (**b**) antioxidant substances, and (**c**) secondary metabolites. Note: All data have two decimal values retained.

**Figure 8 plants-11-02364-f008:**
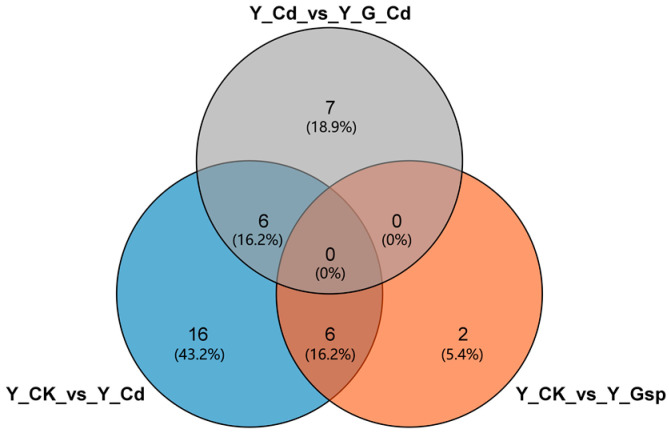
Venn diagram analysis of differential metabolites.

**Figure 9 plants-11-02364-f009:**
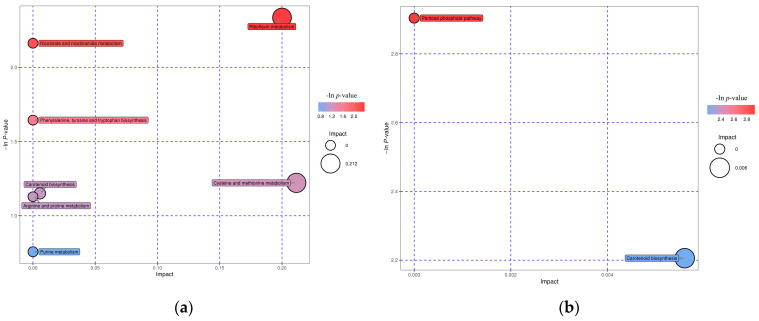
Differential metabolite pathway enrichment. (**a**) Enrichment results of differential metabolites in the CK_vs_Cd group and (**b**) enrichment results of differential metabolites in the Cd_vs_Cd+Gsp group.

**Figure 10 plants-11-02364-f010:**
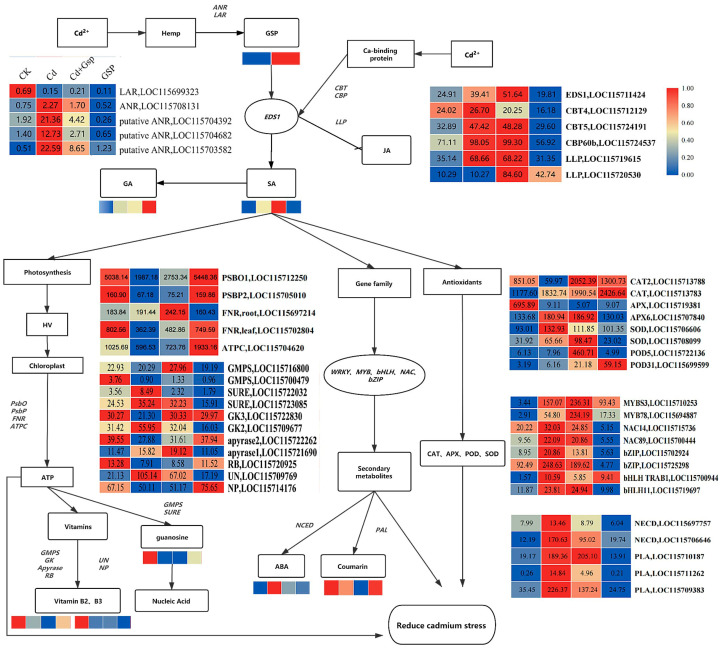
Transcriptome and metabolome joint analyses results. The rectangles in the figure are metabolites or metabolic pathways, and the circles are genes. The heat map below the metabolite is the differential change in the metabolite in the metabolome, and the heat map next to the gene is the differential change in the gene in the transcriptome; the degree of change is set to 0–1. Note: All data have two decimal values retained.

**Figure 11 plants-11-02364-f011:**
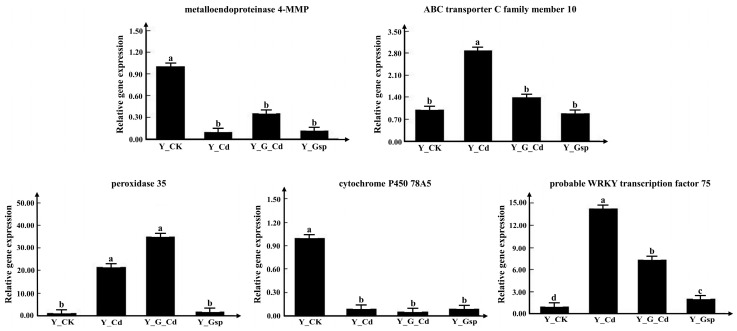
The qPCR verification of transcriptome data. Note: Different letters indicate significant differences in the data under different treatments, *p* < 0.05 (Duncan). Data are expressed as the means (±SD), *n* = 3.

**Figure 12 plants-11-02364-f012:**
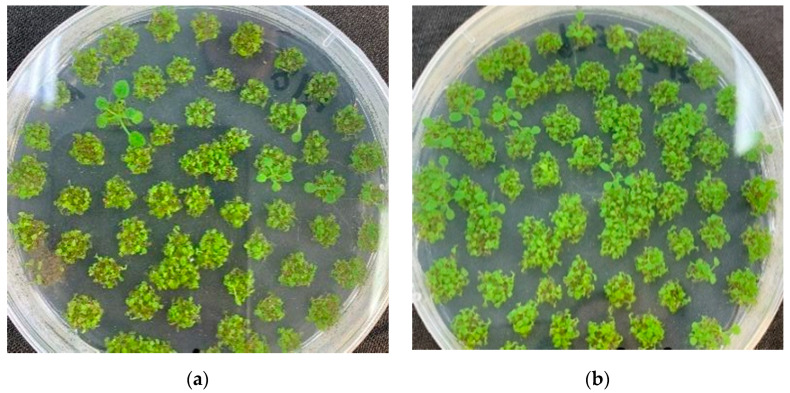
Screening of positive transgenic seedlings in Arabidopsis: (**a**) *CsANR* and (**b**) *CsLAR*.

**Figure 13 plants-11-02364-f013:**
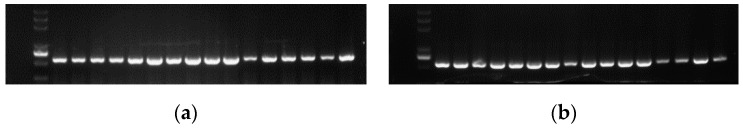
PCR verification of transgenic plants: (**a**) *CsANR* and (**b**) *CsLAR.* PCR detection of hygromycin resistance gene (598 bp), Markers: 5k, 3k, 2k, 1000, 750, 500, 250 bp.

**Figure 14 plants-11-02364-f014:**
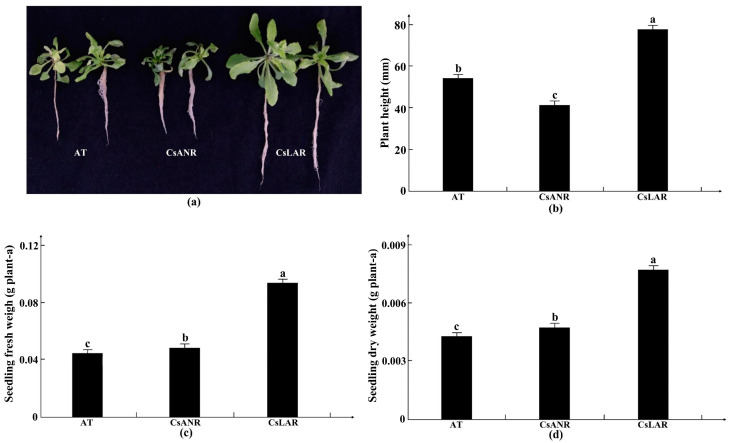
Comparison of different Arabidopsis phenotypes under 100 µmol/L cadmium stress. (**a**) Plant phenotypes of wild-type Arabidopsis vs *CsANR* and *CsLAR* overexpressing Arabidopsis; (**b**) plant height differences; (**c**) fresh weight differences; (**d**) dry weight differences. Note: Different letters indicate significant differences in the data under different treatments, *p* < 0.05 (Duncan). Data are expressed as the means (±SD), *n* = 3.

**Figure 15 plants-11-02364-f015:**
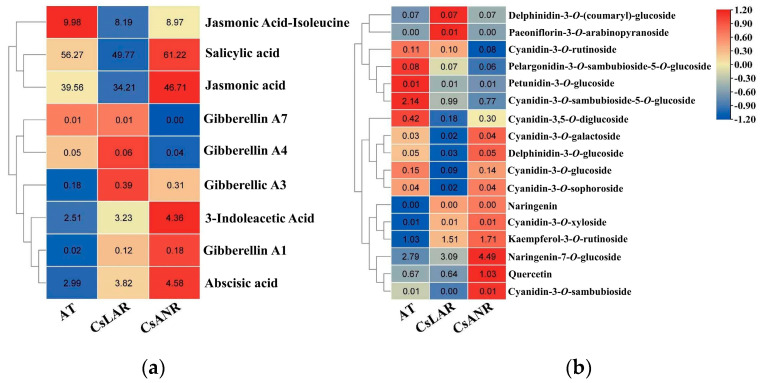
Determination of phytohormones and secondary metabolites in overexpressing Arabidopsis. (**a**) Phytohormone assays in overexpressing Arabidopsis (ng/g); (**b**) determination of secondary metabolites in overexpressing Arabidopsis. The result was rounded to two decimal places.

## Data Availability

The raw data of the RNA-sequencing experiments are available in the NCBI SRA database under accession number PRJNA701120.

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
