# Peer review of "Proanthocyanidins Alleviate Cadmium Stress in Industrial Hemp (Cannabis sativa L.)"

_plants, 2022, doi:10.3390/plants11182364_

Round 1

Reviewer 1 Report

The authors of this manuscript studied the transcriptome and metabolome changes in the industrial hemp caused by proanthocyanidin treatment and Cd stress, to uncover the potential mechanism for proanthocyanidin treatment that alleviate the Cd stress. The manuscript is very poorly written. More importantly, most figures are in very low resolution, which makes it impossible for me to confirm the authors' statements regarding the results. Furthermore, many of their conclusions cannot be supported by their data (check the details in the comments below). For these reasons, I don't recommend this manuscript for publication in its current shape.

1. Fig.1: the resolution of the photo is too low and I can barely read the labels. And the same applies to fig.4, fig.5, fig.6, fig.7, fig.9, fig.10

2. line 165-169: the conclusion that proanthocyanidin treatment alleviate the cadmium stress is already supported by the phenotype difference, but cannot be supported by the data presented here. As the data here only shows the enriched metabolic pathways among the differentially expressed genes. Furthermore, these GO terms (e.g. oxidoreductase activity) represent very general processes, and almost any changes in a growing environment can impact them, so it almost tells nothing about the influence by the proanthocyanidin treatment. Moreover, did authors perform separate GO analysis for the up-regulated and down-regulated genes? What if these processes are only enriched among the genes down-regulated in the pretreated hemp? 

3. line 184-196: this paragraph is very confusing. The authors stated that seven genes were up-regulated but one down-regulated in the pretreated group (line 190-191). Then later they contradicted themselves by saying the proanthocyanidin reduced the expression of most phytohormone genes. Again, because the figure provided was in extremely low resolution, I couldn't even guess the authors' meaning based on the results.

4. line 210-225: MYB, WRKY, and NAC are transcription factor families that could impact lots of biological processes, not just the secondary metabolite biosynthesis. Moreover, the authors only studied the expression of eight transcription factors and none of the genes actually involved in secondary metabolism. The conclusion that the pretreatment represses the expression of most secondary metabolite genes cannot be supported by the data.

5. line 230-235: what are the function of the five genes examined in fig.8? Are they related to the Cd stress?

6. line 307-309: Are ANR, EDS1, CBP, CBT, and LLP the genes evaluated in fig.8? 

7. line 327-329: according to the data presented earlier (line 197-line 209), the pretreatment caused 5 antioxidant genes down-regulated and 5 genes up-regulated as compared to the Cd stressed group. The antioxidant system doesn't appear to be a key component for the pretreatment to alleviate the stress.

Author Response

Dear Reviewer:
    Thank you very much for handling our manuscript entitled “Proanthocyanidins Alleviate Cadmium Stress in Industrial Hemp (Cannabis sativa L.)” (plants-1808552). we are also grateful to you gave invaluable comments to our manuscript. After carefully reviewing your comments, we have edited the images of the submitted articles and changed the text descriptions as well. we also measured the growth indicators of industrial hemp and Arabidopsis thaliana under different treatments by additional experiments, Further verification of the reliability of the article's conclusions. we hope that the improved version will be to your satisfaction. 
The responses to your comments are as following.
Please accept my sincere thanks again.
The best regards
Siqi Huang.
Institute of Bast Fiber Crops, Chinese Academy of Agricultural Sciences
China

Reviewer 2 Report

I have evaluated this manuscript (plants-1808552) entitled “Proanthocyanidins Alleviate Cadmium Stress in Industrial Hemp (Cannabis sativa L.” submitted for publication in ‘Plants’. Topic of manuscript is interesting and falls within the scope of journal. However, in its current form, the manuscript is written and organized poorly and cannot be considered for publication. My main concerns are:

1.       There is no information how the collected data were analyzed statistically. Without rigorous statistical analysis data presented cannot be accepted for publication. What was the design of study? In many Figs, data is presented without any statistical analysis. Moreover, Figs are not self-explanatory, so difficult to follow.

2.       Growth data of hemp and Arabidopsis in quantitative terms is not given. Authors should add data of important growth traits like seedling dry weight and some other important growth-related traits after through statistical analysis. After that growth reduction due to Cd stress can be correlated with antioxidants or other secondary metabolites and physiological indicators. Then the role of over/under expression of certain genes to control above mentioned metabolites can be highlighted.

3.       Some sections need to be rewritten concisely. For example, conclusion section is even longer than the discussion sections of regular manuscripts. Conclusion section should not be more than 4-5 meaningful sentences based on the findings of current study. Data set has almost same trend, so results section can be concised considerably. Avoid results repetition in discussion section and explain results logically. Try to correlate the results rather discussion traits separately.

4.       Expand introduction section to clarify the rationale and need of this study. Divide this section into three paragraphs. In 1st paragraph highlight the problem briefly, in 2nd paragraph discuss various options to sole the issue and in 3rd paragraph explain why you choose this option to sole the issue. Cleary add what is already known information and novel aspect of this study. At the end also add testing hypothesis.

Author Response

Dear Reviewer:

Thank you very much for handling our manuscript entitled “Proanthocyanidins Alleviate Cadmium Stress in Industrial Hemp (Cannabis sativa L.)” (plants-1808552). we are also grateful to you gave invaluable comments to our manuscript. After carefully reviewing your comments, statistical analysis of the data was performed using SPSS software. then, the introduction and conclusions have been revised and edited in the article. We also measured the growth indicators of industrial hemp and Arabidopsis thaliana under different treatments by additional experiments, Further verification of the reliability of the article's conclusions. we hope that the improved version will be to your satisfaction.

The responses to your comments are as following.

Please accept my sincere thanks again.

The best regards

Siqi Huang.

Reviewer 3 Report

The study is very urgent and interesting as the phytoremediation potential of industrial hemp is under consideration due to increasing agricultural areas contaminated by anthropogenic-derived pollutants including heavy metals. However, there are some concerns. 

1. There were no growth analysis conducted. No one morphological parameter was measured. Therefore the description of the effects in 2.1.1 is poor and the conclusion No 1 is not based on the results. 

2. Materials and methods: the method of SOD activity determination is not described

3. Please, provide full latin names of the plant species studied. Why throughout the text in some cases "hemp" is used, but in the others - "cannabis"?

4. Environment setting were the same in the experiments with different stress treatments. Why then "different environments" is used instead of treatments? 

5. What is the reason of growth inhibition by proanthocyanidin treatment?

6. Line 47: What is Nano?

7. Line 55: two genes are not two enzymes.

8. References in the text do not correspond to the requirements (see Author's guidance). 

9. Figures 5-10 require better text quality. 

Author Response

Dear Reviewer:

Thank you very much for handling our manuscript entitled ‘Proanthocyanidins Alleviate Cadmium Stress in Industrial Hemp (Cannabis sativa L.)’ (plants-1808552). we are also grateful to you gave invaluable comments to our manuscript. after carefully reviewing your comments, we have been corrected and explained with the data and analytical results , then, the introduction and conclusions have been revised and edited in the article. We also measured the growth indicators of industrial hemp and Arabidopsis thaliana in different treatments by additional experiments, Further verification of the reliability of the article's conclusions. we hope that the improved version will be to your satisfaction.

The responses to your comments are as following.

Please accept my sincere thanks again.

The best regards

Siqi Huang.

Round 2

Reviewer 2 Report

I have evaluated the revised version of manuscript (plants-1808552) entitled “Proanthocyanidins Alleviate Cadmium Stress in Industrial Hemp (Cannabis sativa L.” submitted for publication in ‘Plants’. In general, the revised draft is in good shape and authors have incorporated the main comments. However, still I have some concerns which needs careful revision.

1.       In Fig 1, label the sub-Figs as a, b, c and d as indicated in captions. Correct the units on x-axis of sub-Fig b, c and d as Plant height (cm), Seedling fresh weigh (g plant-1) and Seedling dry weight (g plant-1). Same is the case with Fig. 14.

2.       Still introduction section is too short and the rationale and need of this study is not clear. Add some literature to highlight the effects of Cd on plant growth and yield, and then move towards solutions. Following articles will be helpful to improve this section:

I.                     Foliar application of seed water extract of Nigella sativa improved maize growth in cadmium-contaminated soil. PLoS ONE 16(7): e0254602 (2021)

II.                   II. Role of ACC-deaminase and/or nitrogen fixing rhizobacteria in growth promotion of wheat (Triticum aestivum L.) under cadmium pollution. Environmental Earth Science 75:267 (2016)

3.       Format references according to journal format both in text and reference list. Scientific names of crops should be in italic both in text and reference list.

Author Response

Dear Reviewer:

Thank you very much for handling our manuscript entitled “Proanthocyanidins Alleviate Cadmium Stress in Industrial Hemp (Cannabis sativa L.)” (plants-1808552). we are also grateful to you gave invaluable comments to our manuscript. After carefully reviewing your comments, we have made changes to the figures(fig.1 and fig.14), add some literature to highlight the effects of Cd on plant growth and yield and solutions,and made formatting changes to the reference format and plant names. In English language and style, we further corrected the deficiencies in the article. we hope that the improved version will be to your satisfaction. The responses to your comments are as following.

Please accept my sincere thanks again.

The best regards

Siqi Huang.

Reviewer 3 Report

All the questions answered

Author Response

Dear Reviewer:

Thank you very much for handling our manuscript entitled ‘Proanthocyanidins Alleviate Cadmium Stress in Industrial Hemp (Cannabis sativa L.)’ (plants-1808552). we are also grateful to you gave invaluable comments to our manuscript. after carefully reviewing your comments, we further corrected the deficiencies in the article about English language and style. we hope that the improved version will be to your satisfaction.

The responses to your comments are as following.

Please accept my sincere thanks again.

The best regards

Siqi Huang.
